# *Fruitless* mutant male mosquitoes gain attraction to human odor

**Nipun S Basrur[1]\*, Maria Elena De Obaldia[1], Takeshi Morita[1], Margaret Herre[1,2], Ricarda K von Heynitz[1†], Yael N Tsitohay[1], Leslie B Vosshall[1,2,3]\***

[1]Laboratory of Neurogenetics and Behavior, The Rockefeller University, New York, United States; [2]Kavli Neural Systems Institute, New York, United States; [3]Howard Hughes Medical Institute, New York, United States

**Abstract** The *Aedes aegypti* mosquito shows extreme sexual dimorphism in feeding. Only females are attracted to and obtain a blood-meal from humans, which they use to stimulate egg production. The *fruitless* gene is sex-specifically spliced and encodes a BTB zinc-finger transcription factor proposed to be a master regulator of male courtship and mating behavior across insects. We generated *fruitless* mutant mosquitoes and showed that males failed to mate, confirming the ancestral function of this gene in male sexual behavior. Remarkably, *fruitless* males also gain strong attraction to a live human host, a behavior that wild-type males never display, suggesting that male mosquitoes possess the central or peripheral neural circuits required to host-seek and that removing *fruitless* reveals this latent behavior in males. Our results highlight an unexpected repurposing of a master regulator of male-specific sexual behavior to control one module of female-specific blood-feeding behavior in a deadly vector of infectious diseases.

**\*For correspondence:**
nbasrur@rockefeller.edu (NSB);
leslie@rockefeller.edu (LBV)

**Present address:** [†] Technical University of Munich, TUM School of Medicine, Munich, Germany

**Competing interests:** The authors declare that no competing interests exist.

## Introduction

Across animals, males and females of the same species show striking differences in behavior. Male *Paradisaeidae* birds-of-paradise perform an elaborate courtship dance to seduce prospective female partners, contorting their bodies in forms resembling flowers, ballerinas, and smiling faces (*Scholes, 2008*). Female *Serromyia femorata* midges pierce and suck conspecific males dry during mating, breaking off his genitalia inside her, thereby supplying the female with both nutrition and sperm (*Edwards, 1920*). Although an astonishing diversity of sexually dimorphic behaviors exists across species, most insights into the genetic and neural basis of sex-specific behaviors have come from a limited set of model organisms (*Matthews and Vosshall, 2020*). Which genes control sexual dimorphism in specialist species that have evolved novel behaviors? Do conserved genes control sexual dimorphism in species-specific behaviors, or do novel genes evolve to control new behaviors?

Many advances in understanding the genetics of sexually dimorphic behaviors have come from the study of *Drosophila melanogaster* fly courtship, where a male fly orients toward, taps, and follows a female fly, extending a wing to produce a courtship song before tasting, mounting, and copulating with her (*Hall, 1994*). Courtship comprises behavioral modules, which are simple discrete behaviors that must be combined to perform a complex behavior and are elicited by different sensory modalities and subsets of *fruitless*-expressing neurons (*Clowney et al., 2015*; *Clyne and Miesenböck, 2008*; *Kohl et al., 2013*; *Ruta et al., 2010*; *von Philipsborn et al., 2011*). Courtship modules include orienting, which is driven by visual information (*Ribeiro et al., 2018*) and persistent following and singing, which are triggered by chemical cues on a female fly (*Clowney et al., 2015*) and guided by vision (*Ribeiro et al., 2018*; *Sten et al., 2020*). The *fruitless* gene is sex-specifically spliced in the brain of multiple insect species including mosquitoes (*Bertossa et al., 2009*; *Gailey et al., 2006*; *Salvemini et al., 2013*) and has been proposed to be a master regulator of male courtship and mating behavior across insects (*Clowney et al., 2015*; *Demir and Dickson,*

**eLife digest** Sexual dimorphism is a phenomenon among animals, insects and plants where the two sexes of a species show differences in body size, physical features or colors. The bushy mane of a male lion, for example, is nowhere to be seen on a female lioness, and only male peacocks have extravagant tails. Most examples of sexual dimorphism, such as elaborate visual displays or courtship behaviors, are linked to mating.

However, there are a few species where behavioral differences between the sexes are not connected to mating. Mosquitoes are an example: while female mosquitoes feed on humans, and are attracted to a person's body heat and odor, male mosquitoes have little interest in biting humans for their blood. Therefore, female mosquitoes are the ones responsible for transmitting the viruses that cause certain blood-borne diseases such as dengue fever or Zika. Determining which genes are linked to feeding behaviors in mosquitoes could allow researchers to genetically engineer females so they no longer bite people, thus stopping the spread of these diseases. Unfortunately, the genes that control mosquito feeding behaviors have not been well studied.

In other insects, some of the genes that control mating behaviors that depend on sex have been identified. For example, a gene called *fruitless* controls courtship behaviors in male flies and silkworms, and is thought to be the 'master regulator' of male sexual behavior across insects. Yet it remains to be seen whether the *fruitless* gene has any effect in mosquitoes, where sex differences relate to feeding habits.

To investigate this, Basrur et al. removed the *fruitless* gene from *Aedes aegypti* mosquitoes. The genetically altered male mosquitoes became unable to mate successfully, but – similar to unmodified males – still preferred sugar water over blood when feeding. Unlike unmodified males, however, the male mosquitoes lacking *fruitless* were attracted to the body odor of a person's arm (like females).

These results reveal that *fruitless*, a gene that controls sex-specific mating behaviors in other insects, controls a sex-specific feeding behavior in mosquitoes. The *fruitless* gene, Basrur et al. speculate, likely gained this role controlling mosquito feeding behavior in the course of evolution. More research is required to fully understand the effects of the *fruitless* gene in male and female mosquitoes.

2005; *Hall, 1994*; *Ryner et al., 1996*; *Seeholzer et al., 2018*; *Tanaka et al., 2017*). Sex-specific splicing of the *fruitless* gene controls several aspects of courtship behavior. Male flies mutant for *fruitless* promiscuously court other males and cannot successfully mate with females (*Ito et al., 1996*; *Ryner et al., 1996*). Forcing male *fruitless* splicing in females triggers orientation and singing behaviors normally only performed by males (*Demir and Dickson, 2005*). In addition, *fruitless* is required for sex-specific aggressive behaviors (*Vrontou et al., 2006*). *Fruitless* encodes a BTB zinc-finger transcription factor that is thought to control cell identity and connectivity during development (*Ito et al., 2016*; *Neville et al., 2014*), as well as the functional properties of neurons in adulthood (*Sethi et al., 2019*). *Fruitless* modulates the expression of a number of potential downstream target genes (*Neville et al., 2014*; *Sato and Yamamoto, 2020*; *Vernes, 2015*) in a cell-type-specific manner (*Brovkina et al., 2020*). Moreover, *fruitless* has a conserved role controlling courtship in multiple *Drosophila* species (*Seeholzer et al., 2018*; *Tanaka et al., 2017*), and sex-specific *fruitless* splicing is conserved across wasps (*Bertossa et al., 2009*) and mosquitoes (*Gailey et al., 2006*; *Salvemini et al., 2013*), suggesting that *fruitless* may act as a master regulator of sexually dimorphic mating behaviors across insects.

Mosquitoes display striking sexually dimorphic mating and feeding behaviors. Only male mosquitoes initiate mating, and only females drink blood, which they require to develop their eggs (*Bowen, 1991*; *Galun et al., 1963*; *Jové et al., 2020*; *Klowden, 1995*). Sexual dimorphism in blood-feeding is one of the only instances of a completely sexually dimorphic feeding behavior because male mosquitoes never pierce skin or engorge on blood. While part of this dimorphism is enforced by sex-specific genitalia (*Spielman, 1964*) or feeding appendages (*Jones and Pilitt, 1973*), there is also a dramatic difference in the drive to hunt hosts between males and female mosquitoes (*Bowen, 1991*; *Roth, 1948*). To blood-feed, females combine multiple behavioral modules

(*Bowen, 1991*). Female *Aedes aegypti* mosquitoes take flight when exposed to carbon dioxide (*Bowen, 1991*; *McMeniman et al., 2014*) and are attracted to human olfactory (*DeGennaro et al., 2013*; *Dekker et al., 2005*; *Zwiebel and Takken, 2004*), thermal, and visual cues (*Liu and Vosshall, 2019*; *McMeniman et al., 2014*; *van Breugel et al., 2015*), and integrate at least two of these cues to orient toward and land on human skin. Engorging on blood is triggered by specific sensory cues tasted by the female (*Galun et al., 1963*; *Jové et al., 2020*). It is not known which genes have evolved to control this unique sexually dimorphic and mosquito-specific feeding behavior. Here, we generate *fruitless* mutant *Aedes aegypti* mosquitoes and show that consistent with observations in *Drosophila*, *fruitless* is required for male mating behavior. Unexpectedly, *fruitless* mutant male mosquitoes gain the ability to host-seek, specifically driven by an attraction to human odor. Our results demonstrate that sexual dimorphism in a single module of a mosquito-specific behavior is controlled by a conserved gene that we speculate has gained a new function in the course of evolution.

## Results

### *Fruitless* is sex-specifically spliced in the mosquito nervous system

We used an arm-next-to-cage assay (*Figure 1A*) to monitor attraction of male and female *Aedes aegypti* mosquitoes to a live human arm. Consistent with their sexually dimorphic blood-feeding behavior, only females were strongly attracted to the arm (*Figure 1B–C*). What is the genetic basis for this extreme sexual dimorphism? We reasoned that *fruitless*, which is alternatively spliced in a sex-specific manner and promotes male courtship and copulation in *D. melanogaster* flies (*Ito et al., 1996*; *Ryner et al., 1996*), may play similar roles in controlling sexually dimorphic behaviors in *Aedes aegypti*. *Fruitless* is a complex gene with multiple promoters and multiple alternatively spliced exons. Downstream promoters drive broadly expressed non-sex-specific *fruitless* transcripts and proteins (*Lee et al., 2000*). A previous study showed (*Salvemini et al., 2013*) and we confirmed that transcripts from the upstream neuron-specific (P1) promoter in the *Aedes aegypti fruitless* gene are sex-specifically spliced (*Figure 1D–H*). Both male and female transcripts include a short male 'm' exon, and female transcripts additionally include a longer female 'f' exon with an early stop codon, predicted to yield a truncated Fruitless protein in the female. However, it is unlikely that any sex-specific Fruitless protein or peptides are stably expressed in adult females. In *Drosophila*, sex-specific female fruitless peptides are not detected (*Lee et al., 2000*), and *transformer* is thought to inhibit translation by binding to female *fruitless* P1 transcripts (*Usui-Aoki et al., 2000*). P1 transcripts of both sexes splice to the first common 'c1' exon, but only male transcripts are predicted to encode full-length Fruitless protein with BTB and zinc-finger domains (*Figure 1E*). By analyzing previously published tissue-specific RNA-seq data (*Matthews et al., 2016*), we verified that of all the *fruitless* exons, only the f exon was sex-specific in *Aedes aegypti* brains (*Figure 1G*). Moreover, while the c1 exon was broadly expressed through downstream promoters, P1 transcripts were specifically expressed in the brain and the antenna, the major olfactory organ of the mosquito (*Figure 1H*), consistent with *fruitless* expression in *Drosophila* (*Stockinger et al., 2005*).

To ask if *fruitless* splicing was conserved across mosquitoes, we sequenced RNA from male and female brains of five different species and assembled de novo transcriptomes for each sex. Three of these species are important arboviral disease vectors because their females blood feed on humans, whereas the two other species only feed on plants (*Bradshaw et al., 2018*; *Zhou et al., 2014*; *Figure 1F*). We identified orthologues of *fruitless* in each species and found that all had conserved 'm' and 'c1' exons and distinct 'f' exons. *Fruitless* was sex-specifically spliced in each of these species with a female-specific 'f' exon and early stop codon, predicted to produce a full-length fruitless protein only in males.

We used CRISPR-Cas9 genome editing (*Kistler et al., 2015*) to disrupt P1 neural-specific *fruitless* transcripts in *Aedes aegypti* to investigate a possible role of *fruitless* in sexually dimorphic mosquito behaviors. We generated two alleles, $fruitless^{\Delta M}$, which introduces a frameshift that is predicted to produce a truncated protein in males, and $fruitless^{\Delta M-tdTomato}$, in which the *fruitless* gene is disrupted by a knocked-in CsChrimson:tdTomato fusion protein (*Figure 1I*). In both alleles, the protein is truncated before the downstream BTB and zinc-finger domains. The $fruitless^{\Delta M-tdTomato}$ line allowed us to visualize cells that express the fluorescent tdTomato reporter under the control of the endogenous *fruitless* regulatory elements. To control for independent background mutations, we used the

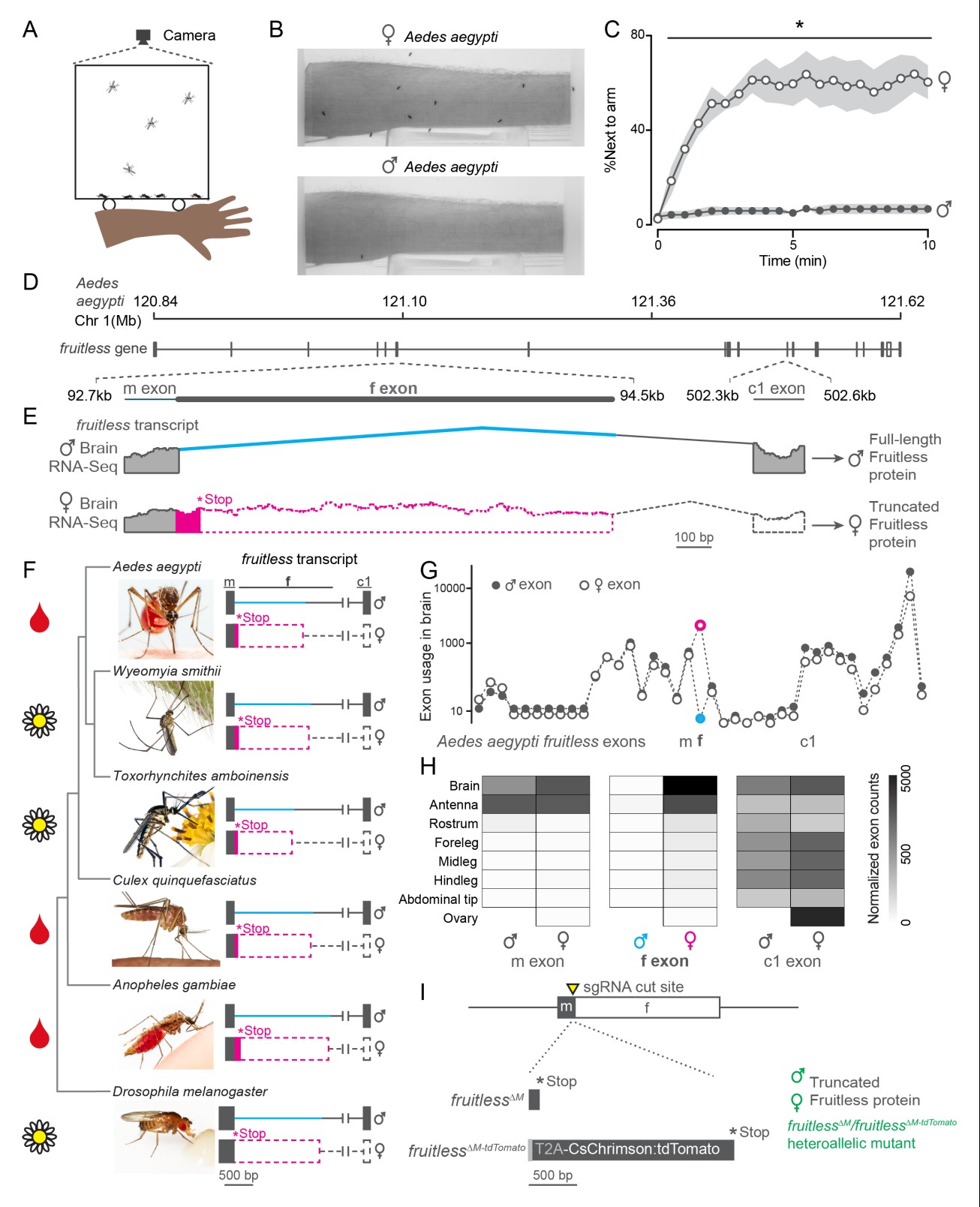

**Figure 1.** Sex-specific mosquito attraction to humans and *fruitless* splicing. (**A, B**) Arm-next-to-cage assay schematic **A** and image **B** with female (top) and male (bottom) *Aedes aegypti* mosquitoes. (**C**) Percent mosquitoes next to arm measured every 30 s. Data are mean ± s.e.m., $n$ = 6 trials, $n$ = 20 mosquitoes/trial; *$p<0.05$, Mann-Whitney test for each time point. (**D, E**) Schematic of *Aedes aegypti fruitless* genomic locus **D** and sex-specific splicing region with RNA-seq read evidence **E**. (**F**) Phylogeny of mosquito species with outgroup *Drosophila melanogaster*, with conserved *fruitless* exon
*Figure 1 continued on next page*

Figure 1 continued

structure inferred from de novo transcriptome assembly. In E, F, coding and non-coding exons are represented by filled and open dashed bars, respectively. *Toxorhynchites rutilus* and *Culex salinarius* images were used to represent *Toxorhynchites amboinensis* or *Culex quinquefasciatus*, respectively. See acknowledgments for photo credits. Cartoons indicate blood-feeding (blood drop) and non-blood-feeding (flower) species. (G, H) *Aedes aegypti fruitless* exon usage based on male and female RNA-seq data (normalized counts) from the indicated tissue plotted for each exon G and m, f, and c1 exons H $n$ = 3–4 independent RNA-seq replicates (*Matthews et al., 2016*). (I) Schematic of generation of *fruitless*$^{\Delta M}$ and *fruitless*$^{\Delta M\text{-}tdTomato}$ mutants.

The online version of this article includes the following source data for figure 1:

**Source data 1.** Source data for *Figure 1*.

heteroallelic *fruitless*$^{\Delta M}$/*fruitless*$^{\Delta M\text{-}tdTomato}$ mutant strain in all subsequent behavior assays (*Figure 1I*). In this heteroallelic mutant, *fruitless* P1 transcripts are disrupted in both males and females. Since full-length fruitless protein is male-specific, we expected that only *fruitless*$^{\Delta M}$/ *fruitless*$^{\Delta M\text{-}tdTomato}$ male mosquitoes would display altered behavioral phenotypes.

## Sexually dimorphic expression of *fruitless* in the mosquito brain and antenna

In *D. melanogaster*, P1 *fruitless* transcripts are expressed in several thousand cells comprising about ~2% of the neurons in the adult brain (*Stockinger et al., 2005*). To examine the distribution of cells expressing *fruitless* in male and female *Aedes aegypti* mosquitoes, we carried out whole mount brain staining to reveal the tdTomato marker expressed from the *fruitless* locus. *Fruitless >tdTomato* is expressed in a large number of cells in both male and female brains (*Figure 2A–F*), as well as in the ventral nerve cord (*Figure 2—figure supplement 1A–B*). *Fruitless >tdTomato* expressing cells innervate multiple regions of the mosquito brain, including the suboesophageal zone, the lateral protocerebral complex, and the lateral horn. These areas have been implicated in feeding (*Jové et al., 2020*), mating (*Seeholzer et al., 2018*), and innate olfactory behaviors (*Datta et al., 2008*) respectively, and also receive projections from *fruitless*-expressing neurons in *Drosophila* (*Seeholzer et al., 2018*; *Stockinger et al., 2005*). The projections of *fruitless >tdTomato* neurons are dramatically sexually dimorphic, with denser innervation in the female suboesophageal zone and the male lateral protocerebral complex (*Figure 2A–F*). We did not detect any gross anatomical differences between heterozygous and heteroallelic *fruitless* mutant male brains or the pattern of *fruitless >tdTomato* expression (*Figure 2B,C,E,F*). We cannot exclude the possibility that there are subtle differences that can only be observed with sparse reporter expression in subsets of cells.

We also examined *fruitless* expression in the periphery. Odors are sensed by olfactory sensory neurons in the mosquito antenna, and each type of neuron projects to a single glomerulus in the antennal lobe of the mosquito brain (*Figure 3A*). We found that, as is the case in *Drosophila* (*Stockinger et al., 2005*), *fruitless >tdTomato* is expressed in olfactory sensory neurons in the antenna of both male and female mosquitoes, and that some of these neurons co-express the olfactory receptor co-receptor Orco (*Figure 3B–E*). *Fruitless >tdTomato* labels a subset of glomeruli in the antennal lobe, with females having about twice as many positive glomeruli compared to males of either genotype (*Figure 3F–L*). There was no difference in the number of *fruitless >tdTomato* labeled glomeruli between wild-type and *fruitless* mutant males (*Figure 3F*), suggesting that *fruitless* does not control sexual dimorphism in the number of glomeruli labeled by *fruitless >tdTomato*.

### *Fruitless* mutant males gain attraction to live human hosts

Given the broad neural expression and sexual dimorphism in *fruitless* circuits, we asked if *fruitless* mutant males showed any defects in sexually dimorphic feeding and mating behaviors. Since only female mosquitoes have the anatomical capacity to pierce skin and artificial membranes (*Jové et al., 2020*; *Klowden, 1995*), we developed a feeding assay in which both females and males are able to feed from warmed liquids through a net without having to pierce a membrane to access the meal (*Figure 4A*). Both wild-type males and females reliably fed on sucrose and did not feed on water. Only wild-type females fed on blood. Even when warm blood was offered and available to males for ready feeding, they still did not find it appetizing (*Figure 4B*). *Fruitless* mutant males fed similarly to

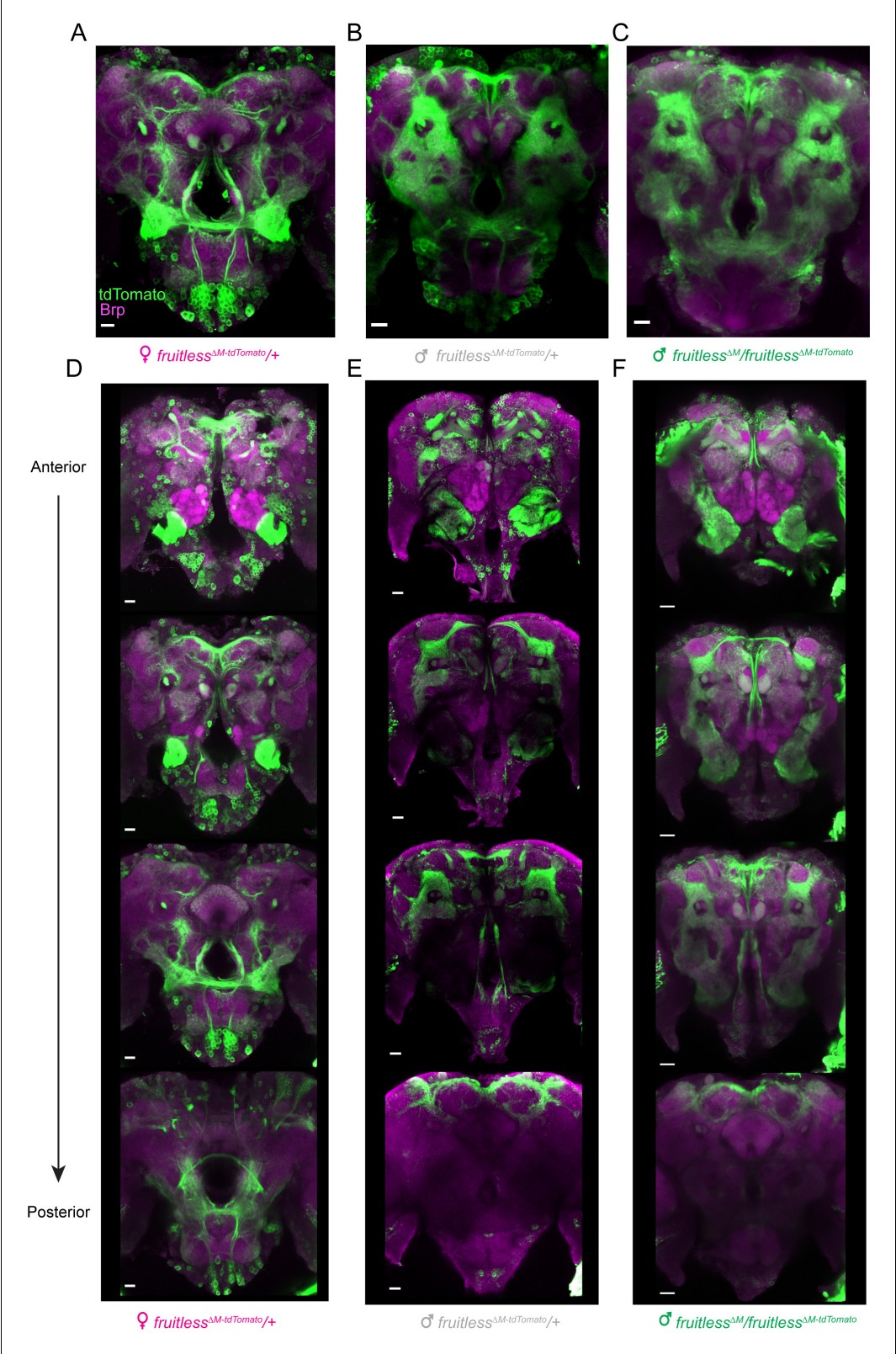

**Figure 2.** Expression of *fruitless* in the mosquito brain. (A–F) Confocal images of brains of the indicated genotypes showing *fruitless >tdTomato* (green) and Brp (magenta) expression. Scale bars, 20 μm. (D–F) Top-to bottom images are optical sections of the same brain, arranged from anterior to posterior.

*Figure 2 continued on next page*

*Figure 2 continued*

The online version of this article includes the following figure supplement(s) for figure 2:

**Figure supplement 1.** Expression of *fruitless* in the female mosquito ventral nerve cord.

their wild-type male counterparts on all meals, suggesting that this behavioral preference is not under the control of *fruitless* in males (*Figure 4B*).

Because *fruitless* plays a key role in male courtship and mating in *Drosophila*, we asked if it is similarly required in *Aedes aegypti*. Since mosquitoes show extremely rapid in-flight mating behavior that is completed in less than 30 s, it is difficult to directly observe or quantify (*Hartberg, 1971*). We used previously developed insemination assays (*Degner and Harrington, 2016*; *Duvall et al., 2017*) to quantify the ability of males to successfully mate (*Figure 4C*). We found that *fruitless* mutant males appeared to contact females but were unable to successfully inseminate wild-type females (*Figure 4D*). This mating failure is consistent with the established role of *fruitless* in *Drosophila* male sexual behavior (*Demir and Dickson, 2005*; *Ryner et al., 1996*).

We then turned to innate olfactory behaviors that govern the search for nectar, which is used as a source for metabolic energy by both males and females, and blood, which is required only by females for egg production. Consistent with the use of these meals, nectar-seeking behavior is not sexually dimorphic, but human host-seeking behavior is sexually dimorphic.

To measure these behaviors, we adapted the Uniport olfactometer (*Liesch et al., 2013*), which is only able to test one stimulus at a time, and developed the Quattroport, an olfactometer that tests attraction to four separate stimuli in parallel (*Figure 4E*). The Quattroport measures both the activation, the participation of the animals in the assay, and attraction, short range attraction to the stimulus (*Figure 4F*). In control experiments, we examined activation responses of wild-type male and females offered a blank, $CO_2$, a human arm, or the floral odor of honey. While males and females showed equivalent activation with a blank and honey, females were more strongly activated to the host-related cues of $CO_2$ and the human arm (*Figure 4G*). To model nectar-seeking behavior, we used honey as a floral odor and glycerol as a control odor as previously described (*Figure 4H*; *DeGennaro et al., 2013*). There was no difference in nectar-seeking as defined by attraction in the Quattroport between wild-type females, males, and *fruitless* mutant males (*Figure 4I*).

We next used the Quattroport with a live human host as a stimulus (*Figure 4J*). As expected, wild-type females robustly and reliably entered traps in response to a live human forearm. In contrast, zero wild-type males entered the trap, consistent with our observations in the arm-next-to-cage assay (*Figure 1A–C*). If *fruitless* function in *Aedes aegypti* were limited to mating and aggression as it is in *Drosophila*, we would expect *fruitless* mutant males to show no interest in a live human host. Unexpectedly, *fruitless* mutant males were as attracted to a live human host as wild-type females (*Figure 4K*). This indicates that *fruitless* males have gained the ability to host-seek, displaying the signature sexually dimorphic behavior of the female mosquito.

## Olfactory and not heat cues attract *fruitless* mutant males to hosts

A live human arm gives off multiple sensory cues that are known to attract female mosquitoes, the most salient of which are body odor and heat. *Fruitless* mutant males might be attracted by heat alone or only the human odor, or to the simultaneous presentation of both cues. To disentangle the contribution of these complex sensory cues to the phenotype we observed, we tested the response of *fruitless* mutant males to each cue in isolation. We first used a heat-seeking assay (*Corfas and Vosshall, 2015*; *McMeniman et al., 2014*) to present heat to mosquitoes in the absence of human odor (*Figure 5A*). Neither *fruitless* mutant nor wild-type males were attracted to the heat cue at any temperature (*Figure 5B*). In contrast, wild-type females showed typical heat-seeking behavior that peaked near human skin temperature (*Figure 5B*).

To ask if *fruitless* mutant males are attracted to human host odor alone, we collected human scent on nylon stockings and presented this stimulus in the Quattroport (*Figure 5C*) to both male and female mosquitoes. Although wild-type males and heterozygous *fruitless* mutant males showed no response to human odor, wild-type females and *fruitless*$^{\Delta M}$/*fruitless*$^{\Delta M-tdTomato}$ females were strongly attracted to human odor (*Figure 5D*). Normal host-seeking in *fruitless* mutant females is

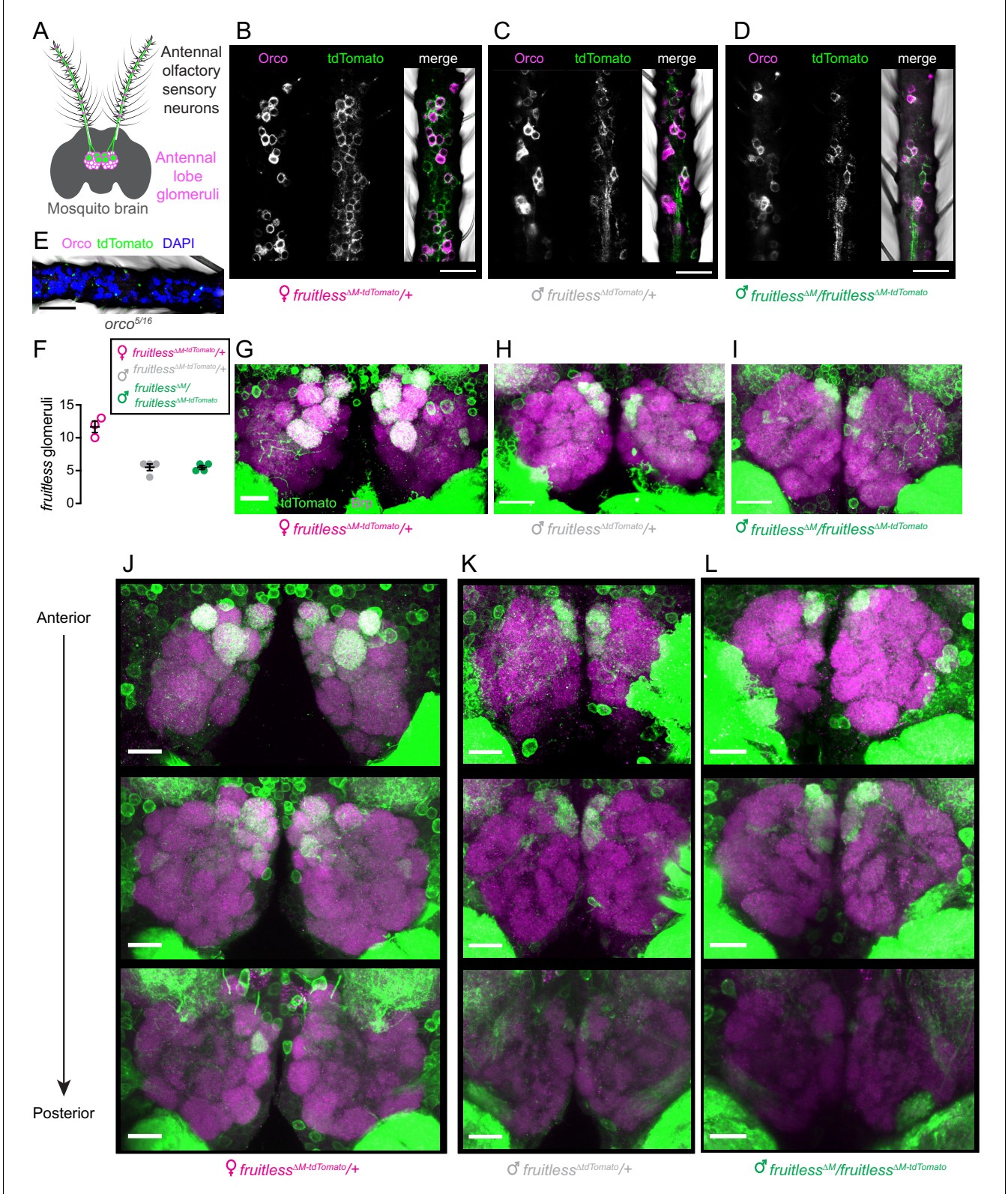

**Figure 3.** Expression of *fruitless* in the mosquito olfactory system. (**A**) Schematic of antennal olfactory sensory neurons and their projections to the antennal lobe of the mosquito brain. (**B–D**) Confocal images of antennae of the indicated genotypes with *fruitless >tdTomato* (green) and Orco (magenta) expression. (**E**) Confocal image of *orco* mutant antenna, as negative control for Orco and tdTomato antibodies, with DAPI (blue). (**F**) Number of antennal lobe glomeruli labeled by *fruitless >tdTomato* in the indicated genotypes. (**G–I**) Confocal images of antennal lobes of the indicated

*Figure 3 continued on next page*

*Figure 3 continued*

genotypes with *fruitless >tdTomato* (green) and Brp (magenta) expression. (J–L) Confocal images of antennal lobes of the indicated genotypes showing *fruitless >tdTomato* (green) and Brp (magenta) expression. Top-to bottom images are optical sections of the same lobes, arranged from anterior to posterior. All scale bars, 20 μm.

The online version of this article includes the following source data for figure 3:

**Source data 1.** Source data for *Figure 3*.

expected since full-length fruitless protein is translated only in males. These females also showed normal blood-feeding, egg-laying, and mating behaviors (*Figure 5—figure supplement 1A–C*), confirming our prediction that *fruitless* acts specifically in the male. We attempted to test the effect of forcing male *fruitless* splicing on female host-seeking (*Figure 5—figure supplement 2A–C*), but found that these animals were inviable due to blood-feeding and egg-laying defects (*Figure 5—figure supplement 2D–I*, *Supplementary file 1*). We then returned to the *fruitless* mutant males. Remarkably, heteroallelic *fruitless* mutant males were strongly attracted to human scent, at levels comparable to wild-type females (*Figure 5D*). These results demonstrate that *fruitless* mutant males have gained a specific attraction to human odor, which drives them to host-seek.

## Discussion

Only female *Aedes aegypti* mosquitoes host-seek, and we have shown that mutating *fruitless* reveals an attraction to human odor in the male mosquito (*Figure 5E*). Previously, *fruitless* was shown to be required for male mating behavior in both *Drosophila* (*Demir and Dickson, 2005*) and *Bombyx* silkmoths (*Xu et al., 2020*). Our work demonstrates that in *Aedes aegypti* mosquitoes, *fruitless* has acquired a novel role in inhibiting female host-seeking behavior in the male (*Figure 5F*). Interestingly, *fruitless* also acts to suppress female-specific aggressive behaviors in male *Drosophila* in addition to its role in promoting male-specific courtship and aggression (*Vrontou et al., 2006*), suggesting a common theme where this gene can repress specific aspects of female-specific behavior. We cannot exclude the possibility that *fruitless* had a broader ancestral role in repressing male-specific host-seeking or feeding but consider this extremely unlikely given the rarity of sexually dimorphic feeding behaviors relative to sexually dimorphic mating behaviors.

Our results suggest that the neural circuits that promote female attraction to human scent are latent in males and suppressed by expression of *fruitless* either during development, or during adulthood. This is in contrast to a model where the ability to host-seek develops exclusively in females. Since males are able to host-seek in the absence of *fruitless*, other components of the sex-determination pathway do not intrinsically regulate the development and function of brain circuits controlling host-seeking behavior, even though this behavior is normally sex-specific. The concept that a latent sex-specific behavior can be revealed by knocking out a single gene was elegantly demonstrated in the mouse (*Mus musculus*). Only male mice court and initiate sexual contact with females and yet knocking out the *Trpc2* gene causes female mice to display these male-specific behaviors (*Kimchi et al., 2007*).

There are field reports of *Aedes aegypti* males being collected near human hosts (*Hartberg, 1971*), which the experimenters interpreted as male *Aedes aegypti* attraction to humans. We note, however, that these field experiments did not control for the presence of females, suggesting that the collected males may have been attracted to the female mosquitoes that attempt to bite humans. In our well-controlled laboratory assays, we were unable to find any evidence of strong attraction to humans in mosquito males at close-range (*Figure 1C*, *Figure 4K*) or at long distances (*Figure 4G*). In these same assays, wild-type male mosquitoes showed strong attraction to floral cues (*Figure 4I*). We cannot exclude that male mosquitoes in the field show some attraction to a human host, but suggest that any attraction in males would be weaker than in wild-type females, or the attraction we demonstrate in *fruitless* mutant males here.

Host-seeking is the first step in a complex sequence of behaviors that lead to blood-feeding. After detecting and flying toward a human host, the female mosquito must land on the human, pierce the skin, and ultimately engorge on blood (*Bowen, 1991*). We have shown that *fruitless* has evolved to control sexual dimorphism in one module of this specialized behavior, the ability to host-seek. Females integrate multiple sensory cues to identify and approach human hosts, and we show

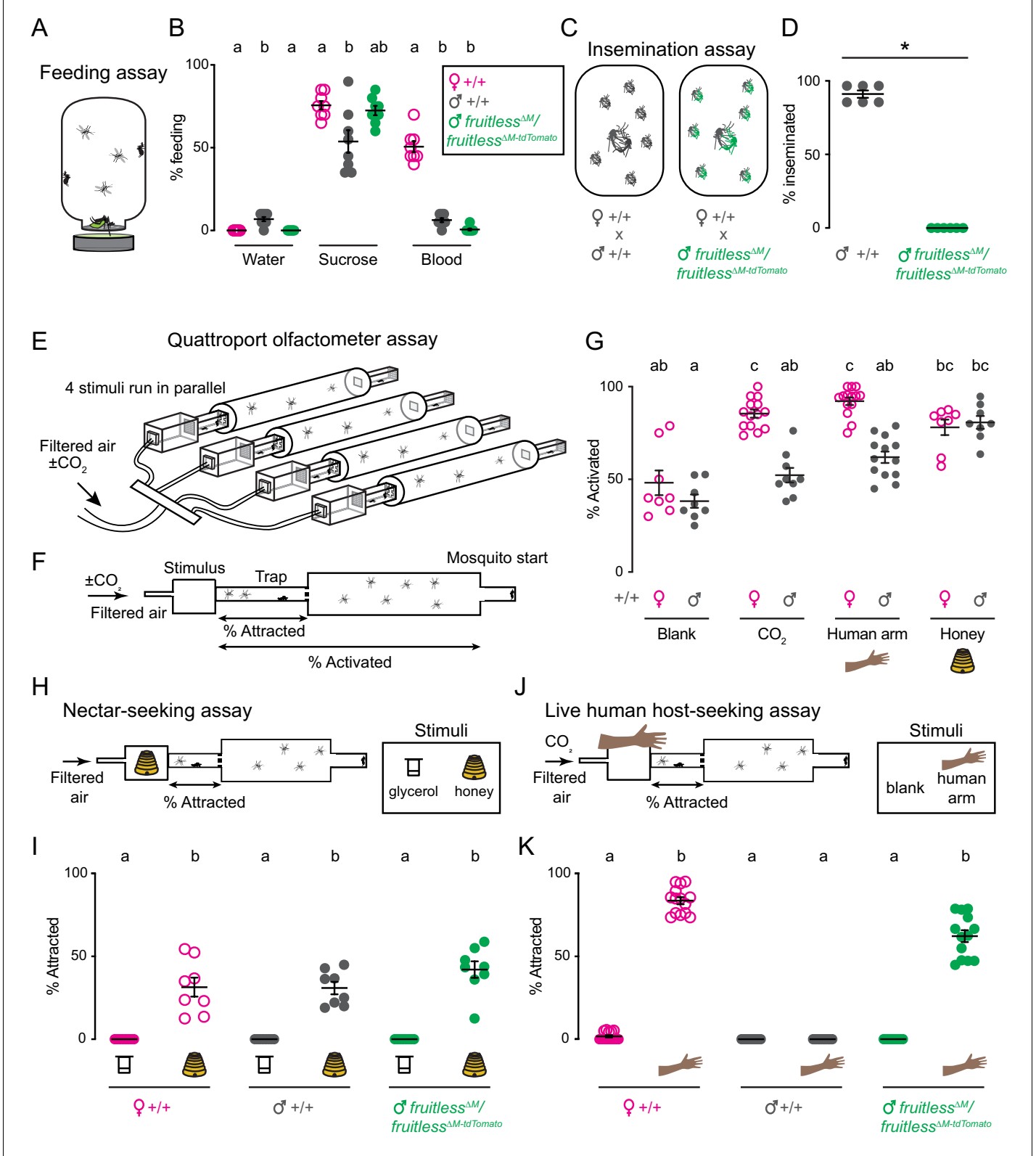

**Figure 4.** Male *fruitless* mutant mosquitoes gain attraction to a live human host. (**A**) Feeding assay schematic. (**B**) Feeding on indicated meal (*n* = 8 trials/meal; *n* = 20 mosquitoes/trial). (**C**) Insemination assay schematic. (**D**) Insemination of wild-type females by males of the indicated genotype (*n* = 6 trials/male genotype, *n* = 20 females/trial; *p=0.0022, Mann-Whitney test). (**E**) Schematic of Quattroport assay, highlighting ability to run multiple stimuli and genotypes simultaneously. (**F**) Side-view schematic of Quattroport, highlighting close-range (attraction) and long-range (activation) metrics (**G**)
*Figure 4 continued on next page*

*Figure 4 continued*

Percent activated animals, *n* = 8–14 trials/group, *n* = 17–28 mosquitoes/trial. (H, J) Quattroport assay schematic for nectar-seeking H and live human host seeking J. 1% $CO_2$ is added to the airstream in the live human host seeking assay J. (I, K) Percent of attracted animals (*n* = 8–14 trials per group, *n* = 17–28 mosquitoes/trial). Data in B, D, G, I, K are mean ± s.e.m. In B, I, G, K, data labeled with different letters are significantly different from each other (Kruskal-Wallis test with Dunn's multiple comparisons, p<0.05). In B, comparisons are made between genotypes for each meal. In I, K, comparisons are made between all genotypes and stimuli.

The online version of this article includes the following source data for figure 4:

**Source data 1.** Source data for *Figure 4*.

that *fruitless* controls the response to just one of those cues, human odor. Sexual dimorphism in thermosensation, or in subsequent feeding behaviors does not appear to be controlled by *fruitless* in the male mosquito, since neither wild-type nor mutant *fruitless* males will drink warm blood. The modular genetic organization of mosquito behavior sparks intriguing parallels to other complex sexually dimorphic behaviors like mouse parenting (*Kohl et al., 2018*). To be effective parents, female mice must build nests, and then retrieve, groom, and nurse their pups. In the deer mouse *Peromyscus*, the conserved peptide vasopressin has evolved to control nest building (*Bendesky et al., 2017*). In both *Peromyscus* and *Aedes*, a conserved gene has gained control over a single aspect of a complex behavior. It has been hypothesized that certain classes of genes like neuromodulators and transcription factors are more likely to underlie phenotypic differences between species (*Bendesky and Bargmann, 2011*; *Martin and Orgogozo, 2013*; *Tosches, 2017*), and our study demonstrates that this is true even for an entirely novel behavior.

Where in the nervous system is *fruitless* required to suppress host seeking in male mosquitoes? *fruitless* might function in the antenna to modulate the detection of human odor in male mosquitoes, perhaps by tuning the functional or anatomical properties of olfactory sensory neurons. Such a role has been recently demonstrated in the aging-dependent sensitization of the male *Drosophila* antennal response to pheromones (*Sethi et al., 2019*; *Zhang and Su, 2020*; *Zhao et al., 2020*). Alternatively, both wild-type males and females might detect human odor, and *fruitless* could function in the central brain to reroute these signals to drive different motor outputs, as has been demonstrated with the sexually dimorphic response to *Drosophila* pheromones (*Datta et al., 2008*; *Kohl et al., 2013*; *Ruta et al., 2010*). To distinguish between these two models, we would require significant advances in technology and mosquito genetics, including a *fruitless* driver line to image neural responses to human odor. Despite significant effort, we were unable to generate a viable *fruitless* driver line both because of tight genetic linkage of *fruitless* to the sex-determining M locus (*Hall et al., 2015*) and because gene-targeted females failed to blood-feed and were therefore sterile (*Supplementary file 1*). To explore central brain *fruitless*+ circuits, we would need to be able to subset expression to label and drive reporters or rescue *fruitless* expression in sparse populations of neurons, a technology that is still out of reach. Advances in mosquito genetic tools, such as the successful implementation of orthogonal transcriptional activator reagents, combined with sparse labeling approaches will be required to gain mechanistic insight into *fruitless* function within mosquito host-seeking circuits. We note that these advances were not trivial in *D. melanogaster*, requiring efforts from multiple laboratories over the past decade (*Datta et al., 2008*; *Kohl et al., 2013*; *Ruta et al., 2010*), and expect that the generation of these tools and the subsequent characterization of the circuit will be significantly more challenging in the mosquito, a non-model organism.

Our work suggests that *fruitless* has evolved the novel function of enforcing female-specific host-seeking while maintaining its presumably ancestral male mating function. How might *fruitless* have evolved to control host-seeking? One possibility is that non-sex-specific host-seeking neural circuits first emerged in the ancestral mosquito, and then secondarily began to express *fruitless* to suppress the development or adult function of host-seeking circuits specifically in males. Another possibility is that mosquitoes duplicated and co-opted the ancestral *fruitless*-expressing mating neural circuits and retuned the inputs and outputs into this circuit to drive host-seeking. Circuit duplication is one of the mechanisms by which neural circuits are proposed to evolve (*Tosches, 2017*) and has been demonstrated in the case of vocal learning (*Chakraborty and Jarvis, 2015*) and in the evolution of cerebellar nuclei (*Kebschull et al., 2020*). In these duplicated mosquito circuits, *fruitless* function would have switched from promoting mating to inhibiting host-seeking in males. We speculate that both possibilities allow for *fruitless* to control both sex-specific host-seeking and mating behaviors,

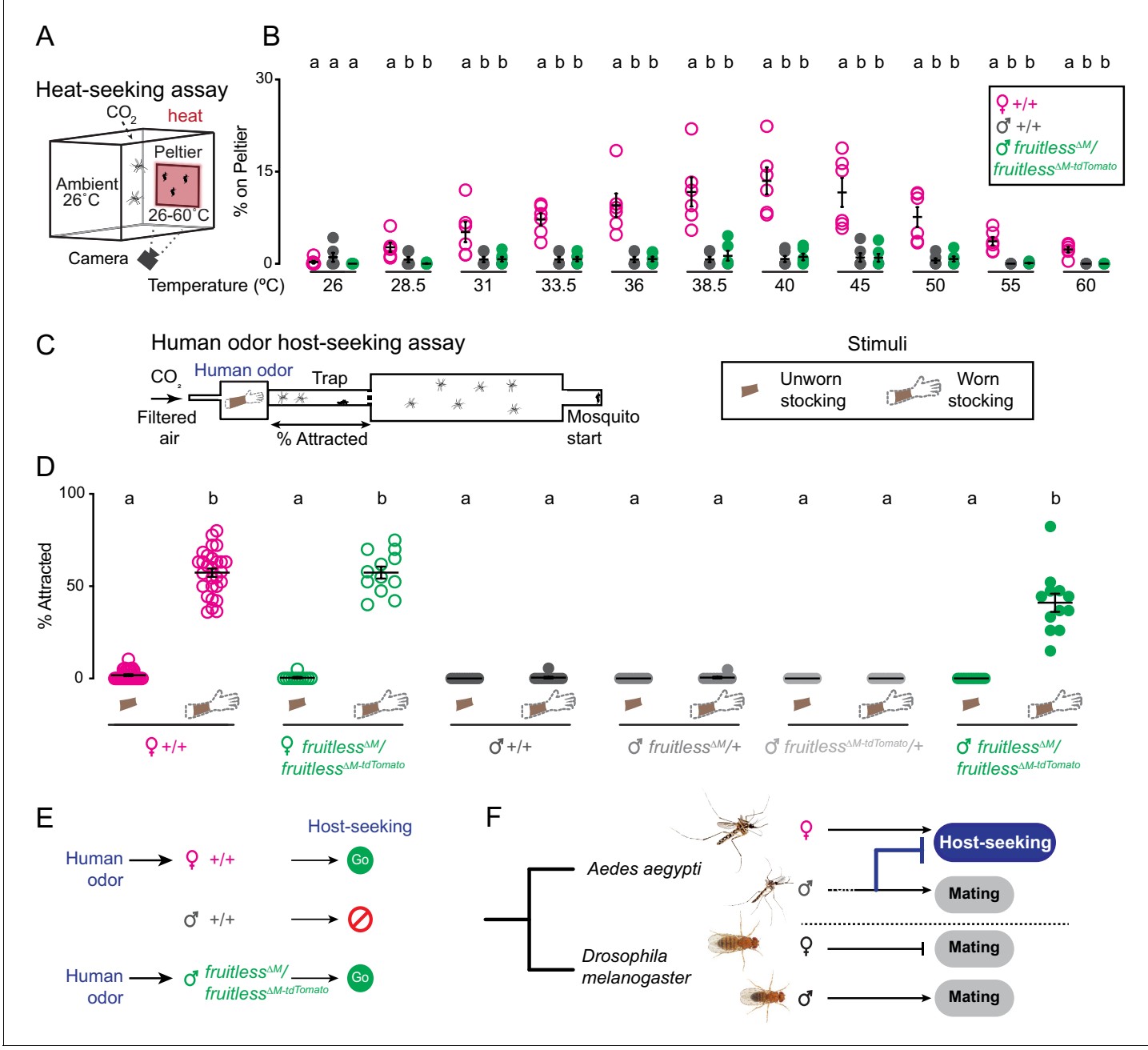

**Figure 5.** Olfactory cues selectively drive male *fruitless* mutant attraction to humans. (**A**) Heat-seeking assay schematic. A 20 s pulse of 10% $CO_2$ is added to the assay. (**B**) Percent of animals on Peltier. Data are mean ± s.e.m., *n* = 6 trials/temperature, *n* = 50 mosquitoes/trial. Data labeled with different letters are significantly different from each other, within each temperature. (**C**) Schematic of human odor host-seeking assay (left) and stimuli (right). (**D**) Percent of attracted animals. Data are mean ± s.e.m., *n* = 8–14 trials/group, *n* = 17–28 mosquitoes/trial. (**E–F**) Summary of results and model of gain of *fruitless* function in *Aedes aegypti*. Photo credit: *Aedes aegypti* (Alex Wild); *D. melanogaster* (Nicolas Gompel). In B, D, data labeled with different letters are significantly different from each other (Kruskal-Wallis test with Dunn's multiple comparisons, p<0.05). In B, comparisons are made between genotypes at each temperature. In D, comparisons are made across all genotypes and stimuli.

The online version of this article includes the following source data and figure supplement(s) for figure 5:

**Source data 1.** Source data for *Figure 5*.
**Figure supplement 1.** No significant blood-feeding or mating defects in *fruitless*$^{\Delta M}$ females.
**Figure supplement 1—source data 1.** Source data for *Figure 5—figure supplement 1*.
**Figure supplement 2.** Female *fruitless*$^{\Delta F}$ mutant mosquitoes have blood-feeding and egg-laying defects.
**Figure supplement 2—source data 1.** Source data for *Figure 5—figure supplement 2*.

and identification and molecular profiling of the *fruitless* cells controlling host-seeking and mating will help distinguish between these models. Our work highlights *fruitless* as a potential means to investigate the circuit basis of *Aedes aegypti* host seeking, a behavior that is responsible for infecting millions of people with life-threatening pathogens.

# Materials and methods

## Key resources table

| Reagent type (species) or resource | Designation | Source or reference | Identifiers | Additional information |
|---|---|---|---|---|
| Gene (*Aedes aegypti*) | *fruitless* | RefSeq | LOC5567734 | |
| Gene (*Anopheles gambiae*) | *fruitless* | VectorBase | AGAP000080 | |
| Gene (*Culex quinquefasciatus*) | *fruitless* | RefSeq | CPIJ010853 | |
| Gene (*Wyeomyia smithii*) | *fruitless* | This paper | | *Figure 1—source data 1* |
| Gene (*Toxorhynchites amboinensis*) | *fruitless* | This paper | | *Figure 1—source data 1* |
| Strain, strain background (*Aedes aegypti*, both sexes) | LVP-IB12 | BEI resources | MRA-735 | |
| Strain, strain background (*Anopheles gambiae*, both sexes) | G3 | Flaminia Catteruccia (Harvard University) | | |
| Strain, strain background (*Culex quinquefasciatus*, both sexes) | JHB | BEI resources | NR-43025 | |
| Strain, strain background (*Wyeomyia smithii*, both sexes) | PB | William Bradshaw and Christina Holzapfel (University of Oregon) | | |
| Strain, strain background (*Toxorhynchites amboinensis*, both sexes) | | Laurence Zwiebel (Vanderbilt University) | | |
| Genetic reagent (*Aedes aegypti*, both sexes) | *fruitless*$^{\Delta M}$ | This paper | | *fruitless* mutant strain; eggs available on request |
| Genetic reagent (*Aedes aegypti*, both sexes) | *fruitless*$^{\Delta M\text{-}tdTomato}$ | This paper | | *fruitless* mutant strain; eggs available on request |
| Antibody | Mouse monoclonal anti-*Apocrypta bakeri* Orco antibody | Vanessa Ruta (The Rockefeller University) (PMID:30111839) | 15B2 | IF(1:50) |
| Antibody | Rabbit polyclonal anti-RFP antibody | Rockland | Cat# 600-401-379, RRID:AB_2209751 | IF(1:200) |
| Antibody | Mouse monoclonal anti-Brp antibody | DSHB | Cat# nc82, RRID:AB_2314866 | IF(1:5000) |
| Antibody | Goat polyclonal anti-mouse Alexa Fluor 488 antibody | Thermo Fisher | Cat# A-11001, RRID:AB_2534069 | IF(1:500) |

*Continued on next page*

*Continued*

| Reagent type (species) or resource | Designation | Source or reference | Identifiers | Additional information |
|---|---|---|---|---|
| Antibody | Goat polyclonal anti-mouse Alexa Fluor 555 Plus antibody | Thermo Fisher | Cat# A32732, RRID:AB_2633281 | IF(1:500) |
| Antibody | Goat polyclonal anti-mouse Alexa Fluor 647 polyclonal antibody | Thermo Fisher | Cat# A-21235, RRID:AB_2535804 | IF(1:500) |
| Recombinant DNA reagent | FruitlessΔ-T2A-QF2 (plasmid) | This paper | | Plasmid used to generate *fruitless*$^{ΔM}$ strain |
| Recombinant DNA reagent | FruitlessΔ-T2A-CsChrimson-Td Tomato (plasmid) | This paper | | Plasmid used to generate *fruitless*$^{ΔM-tdTomato}$ strain |
| Commercial assay or kit | NEBuilder HiFi DNA Assembly | NEB | NEB:E5520S | |
| Commercial assay or kit | HiScribe Quick T7 kit | NEB | NEB:E2050S | |
| Peptide, recombinant protein | Cas9 | PNABio | PNABio:CP01-200 | |
| Sequence-based reagent | sgRNA_for | This paper | PCR primer for sgRNA | GAAATTAATACGACTCACTATAGCACCGAAGGTATGTTGAGGTTTTAGAGCTAGAAATAGC |
| Sequence-based reagent | sgRNA_rev | This paper | PCR primer for sgRNA | AAAAGCACCGACTCGGTGCCACTTTTTCAAGTTGATAACGGACTAGCCTTATTTTAACTTGCTATTTCTAGCTCTAAAAC |
| Software, algorithm | STAR v 2.5.2a | | | RNAseq alignment |
| Software, algorithm | Trinity v 2013-03-25 | | | De novo assembly |
| Software, algorithm | Prism v8 | GraphPad | Prism 8 | statistics |
| Software, algorithm | MacVector v15.0.3 | MacVector | MacVector | Plasmid cloning |
| Other | DAPI stain | Sigma | D9542 | 1:10,000 |
| Other | Quattroport | This paper | | https://github.com/VosshallLab/Basrur_Vosshall2020; *Basrur, 2021*; (copy archived at swh:1:rev:529b0ba393bac09207d298e73e425b452338e876) |

## Mosquito rearing and maintenance

*Aedes aegypti* wild-type laboratory strains (Liverpool-IB12) were maintained and reared at 25–28°C, 70–80% relative humidity with a photoperiod of 14 hr light: 10 hr dark (lights on at 7 a.m.) as previously described (*DeGennaro et al., 2013*). All behavioral assays were performed at these conditions of temperature and humidity. Adult females were blood-fed on mice for stock maintenance and on human subjects for initial stages of mutant generation. *Anopheles gambiae* (G3 strain), *Wyeomyia smithii* (PB strain), *Toxorhynchites amboinensis*, and *Culex quinquefasciatus* (JHB strain) were reared in similar conditions, following previously described protocols for each species (*Bradshaw et al., 2018*; *Werling et al., 2019*; *Zhou et al., 2014*). Adult mosquitoes of each species were provided constant access to 10% sucrose.

## RNA-sequencing

Seven- to 14-day-old mosquitoes of each species were cold-anesthetized and kept on ice for up to 1 hr or until dissections were complete. Brains were dissected in ice-cold RNase-free phosphate-buffered saline (PBS) (Invitrogen AM9625) on ice, moved into a microfuge tube with forceps, and immediately snap frozen in a cold block (Simport S700-14) chilled to −80°C on dry ice. Each sample group was dissected in parallel to avoid artefacts and batch effects, and five brains were used per sample. Dissected tissue was stored at −80°C until RNA extraction. RNA extraction was performed using the

PicoPure Kit (ThermoFisher #KIT0204) following the manufacturer's instructions, including DNase treatment. Samples were run on a Bioanalyzer RNA Pico Chip (Agilent #5067–1513) to determine RNA quantity and quality. Libraries were prepared using the Illumina TruSeq Stranded mRNA kit #20020594, following manufacturer's instructions. Library quantity and quality were evaluated using High Sensitivity DNA ScreenTape Analysis (Agilent #5067–5585) prior to pooling. Barcoded samples from all non-*Aedes* tissues were pooled in an equal ratio before distributing the pool across two sequencing lanes. Sequencing was performed at The Rockefeller University Genomics Resource Center on a NextSeq 500 sequencer (Illumina). All reads were 2 × 150 bp. Data were de-multiplexed and delivered as fastq files for each library. Sequencing reads have been deposited at the NCBI Sequence Read Archive (SRA) under BioProject PRJNA612100.

### *Fruitless* splicing analysis

Reads from individual *Aedes* libraries were mapped to the AaegL5 genome (*Matthews et al., 2018*) using STAR version 2.5.2a with default settings (*Dobin et al., 2013*). Raw counts were used for differential splicing analysis in *Aedes aegypti* using DEXSeq version 1.32.0 (*Anders et al., 2012*) as per author instructions. For the other mosquito species without genomes or incomplete genome annotations, we assembled sex-specific de novo transcriptomes using Trinity version 2013-03-25 with default settings (*Grabherr et al., 2011*). We then searched for *fruitless* orthologues in each species using BLAST 2.6.0 (*Altschul et al., 1990*), aligned hits to *Aedes aegypti fruitless* P1 transcripts using MacVector version 15.0.3, and picked the best match for each exon, species, and sex. *Fruitless* exon sequences are found in *Figure 1—source data 1*.

### *Fruitless*$^{\Delta M}$ and *fruitless*$^{\Delta M\text{-}tdTomato}$ strain generation

The *fruitless* gene was targeted using CRISPR-Cas9 methods as previously described (*Kistler et al., 2015*). Gene-targeting reagents were injected into wild-type Liverpool-IB12 embryos at the Insect Transformation Facility at the University of Maryland Institute for Bioscience and Biotechnology Research. For each line, either 2000 or 1000 embryos were injected with 600 ng/μL plasmid, 300 ng/μL Cas9 protein (PNABio CP01-200), and 40 ng/μL sgRNA. Proper integration was confirmed in each strain using polymerase chain reaction (PCR) and sequencing. Animals were then back-crossed to wild-type Liverpool-IB12 for at least four generations.

All homology arms for homology-directed integration were isolated by PCR using Liverpool-IB12 genomic DNA. sgRNA DNA template was prepared by annealing oligonucleotides as previously described (*Kistler et al., 2015*). In vitro transcription of sgRNA template was performed using HiScribe Quick T7 kit (New England Biolabs #E2050S) following the manufacturer's directions and incubating for 4 hr at 37°C. Following transcription and DNAse treatment for 15 min at 37°C, sgRNA was purified using Ampure RNAse-free SPRI beads (Beckman-Coulter #A63987) and eluted in Ultra-pure water (Invitrogen #10977–015). For all plasmids, fragments were generated by PCR from the indicated template with the indicated primers (*Supplementary file 2*) and assembled using NEBuilder HiFi DNA Assembly (NEB E5520S). Plasmids were transformed into NEB competent cells (NEB C2987I), purified with the NucleoBond Xtra Midi Endotoxin Free kit (Clontech 740420.50), and sequence verified.

The *fruitless*$^{\Delta M}$ mutant was generated in the course of attempting to generate a *fruitless* QF2 knock-in mutant (*Supplementary file 1*) (see below). One of the families had viable 3xP3-dsRed-positive offspring and an out-of-frame QF2 insertion, which was predicted to produce a truncated fruitless protein in males. This was the *fruitless*$^{\Delta M}$ mutant allele we used in the study.

The *fruitless*$^{\Delta M\text{-}tdTomato}$ knock-in/knock-out strain was generated by inserting a cassette containing T2A followed by CsChrimson fused to the fluorescent protein tdTomato and the 3xP3-EYFP strain marker. We obtained two independent viable lines and selected one for use in this study. We used the CsChrimson:tdTomato protein expressed from the *fruitless* locus in *fruitless*$^{\Delta M\text{-}tdTomato}$ animals as a marker for *fruitless* expression in these studies.

CsChrimson is a red-light-activated cation channel, and we originally generated this animal with the intention of optogenetically manipulating behavior. However, CsChrimson:tdTomato intrinsic fluorescence was not visible under a confocal microscope, even at high laser intensities, and required immunofluorescent amplification in all our images. When animals were fed with retinal, the necessary co-factor which was absent in all other experiments, and we attempted to substitute human odor

and $CO_2$ with optogenetic activation of *fruitless*+ neurons in a blood-feeding assay, we did not observe increased feeding or any other behaviors (preliminary data not shown). Although we were unable to see evidence of CsChrimson activity in these optogenetic experiments, potential background levels of CsChrimson-driven activation of *fruitless*-expressing neurons is an important concern to rule out when considering the behavioral data in this paper. We note that animals were not fed trans-retinal, the necessary cofactor for CsChrimson activity. *Fruitless*$^{\Delta M\text{-}CsChrimson\text{-}tdTomato/+}$ males and females were able to mate normally, and *fruitless*$^{\Delta M\text{-}CsChrimson\text{-}tdTomato/+}$ females show normal blood-feeding and egg-laying behavior (*Figure 5—figure supplement 1A–C*). We have not examined *fruitless*$^{\Delta M\ /\Delta M}$ animals due to the difficulty of obtaining these animals without molecular genotyping of each individual. However, given the weak expression of CsChrimson and the robust behavior of heterozygote animals, we consider it unlikely that this allele is significantly affecting mosquito behavior.

## Attempted generation of *fruitless* QF2/QF2w knock-in mutants

We attempted to generate *fruitless* P1-specific driver lines by knocking in a cassette containing the ribosomal-skipping peptide T2A followed by the transcriptional activator QF2 (*Riabinina et al., 2015*), with 3xP3-dsRed as an insertion marker as previously described (*Matthews et al., 2019*). In this knock-in/knock-out strain, we aimed to disrupt the *fruitless* gene as well as generate a driver line that would allow us to label and manipulate *fruitless*-expressing neurons. We recovered seven independent 3xP3-dsRed positive G1 families. However, all females with one copy of the correct integration did not blood-feed after many attempts using multiple different human hosts. Males with one copy of this insertion did not mate with wild-type females. Since blood-meals are required for *Aedes aegypti* egg-development, this line could not be maintained. We next tried to knock-in the weaker QF2w transcriptional activator, and recovered six independent families, all of which showed the same blood-feeding and mating defects (*Supplementary file 1*). We speculate that toxicity of QF2 or QF2w may affect the function or viability of *fruitless*-expressing neurons, leading to the behavioral defects we observed. The cause of Q-system toxicity, even attenuated from Q to QF2 to QF2w is unknown (*Riabinina et al., 2015*). We speculate that this toxicity is unrelated to the *fruitless* locus, because *fruitless*$^{\Delta M/+}$ animals had no phenotype as heterozygotes, unlike *fruitless*$^{\Delta QF2/+}$ and *fruitless*$^{\Delta QF2w/+}$ animals.

## Attempted generation of sex-switched *fruitless* mutants

To ask if fruitless protein was sufficient to inhibit host-seeking behavior in females, we attempted to force females to express male fruitless protein by deleting the female exon of *fruitless* P1 transcripts and forcing male *fruitless* splicing in female brains (*Supplementary file 1*). For the *fruitless*$^{\Delta F}$ line, embryos were injected with 300 ng/µL Cas9 protein, 125 ng/µL oligonucleotide with template repairing the splice site, 40 ng/µL each of two sgRNAs targeting the beginning and end of the female-specific exon. We recovered multiple G1 animals with the correct integration, as verified by PCR and sequencing. Male *fruitless* splicing in *fruitless*$^{\Delta}$ females was verified with reverse-transcription PCR (data not shown). G2 *fruitless*$^{\Delta F}$ females did not fully blood-feed or lay eggs even though they were successfully inseminated by wild-type males (*Figure 5—figure supplement 2D–I*). It was therefore impossible to maintain these lines. We do not know if the blood-feeding defect was due to a failure to respond to the host or some other behavioral or anatomical defect. Since *fruitless* is tightly linked to the male-determining locus, it was not an option to maintain this targeted allele in males. Integrations on the male chromosome would yield ~1/500 females with the recombinant allele, and integrations on the female chromosome yield inviable females and rare recombinant males. In either scenario, the *fruitless*$^{\Delta F}$ insertion is unmarked and would need to be followed by PCR genotyping. We also attempted to generate a line where we both deleted the female *fruitless* exon and knocked-in an intronic 3xP3 fluorescent marker, which would allow us to maintain this line in males and use the marker to select rare recombinants for behavioral analysis. However, females with this integration did not have any behavioral phenotypes, suggesting that the intronic 3xP3 marker interfered with regular *fruitless* splicing in both males and females. These difficulties precluded any further investigation of the phenotype of expressing full-length fruitless protein in females.

## Antibody staining – brain whole mounts

Dissection of adult brains and immunostaining was carried out as previously described (*Jové et al., 2020*; *Matthews et al., 2019*). Six- to 14-day-old mosquitoes were anesthetized on ice and decapitated. Heads were fixed in 4% paraformaldehyde (Electron Microscopy Sciences 15713 s), 1X Ca$^{+2}$, Mg$^{+2}$ free PBS (Thermo 14190144), 0.25% Triton X-100 (Sigma 93443), and nutated for 3 hr at 4°C. Brains were then dissected and placed in cell-strainer caps (Falcon #352235) in a 24-well plate. All subsequent steps were performed on a low-speed orbital shaker. Brains were washed for 15 min at room temperature in 1x PBS with 0.25% Triton X-100 (0.25% PBT) at least six times. Brains were permeabilized with 4% Triton X-100 with 2% normal goat serum (Jackson ImmunoResearch #005-000-121) in PBS at 4°C for 2 days. Brains were washed for 15 min at least six times with 0.25%PBT at room temperature. Brains were incubated in 0.25% PBT plus 2% normal goat serum with primary antibodies at the following dilutions: rabbit anti-RFP (Rockland 600-401-379) 1:200 and mouse anti-*Drosophila melanogaster* Brp (nc82) 1:5000. The nc82 hybridoma developed by Erich Buchner of Universitätsklinikum Würzburg was obtained from the Developmental Studies Hybridoma Bank, created by the NICHD of the NIH and maintained at The University of Iowa, Department of Biology, Iowa City, IA 52242. Primary antibodies were incubated for 2 nights at 4°C and then washed at least six times for 15 min with 0.25% PBT at room temperature. Brains were incubated with secondary antibody for 2 nights at 4°C with secondary antibodies at 1:500% and 2% normal goat serum. Secondary antibodies used were goat anti-rabbit Alexa Fluor 555 (Thermo A32732) and goat anti-mouse Alexa Fluor 647 (Thermo A-21235). Brains were then washed for 15 min at least six times with 0.25% PBT at room temperature and mounted in Slowfade Diamond (Thermo S36972) using #1.5 coverslips as spacers before confocal imaging.

## Antibody staining – antennal whole mounts

This protocol was adapted from a *Drosophila* embryo staining protocol (*Manning and Doe, 2017*). Six- to 10-day-old mosquitoes were anesthetized, decapitated, and placed in 1.5 mL 5 U/mL chitinase (Sigma C6137) and 100 U/mL chymotrypsin (Sigma CHY5S) in 119 mM NaCl, 48 mM KCl, 2 mM CaCl2, 2 mM MgCl2, 25 mM HEPES buffer on ice. Male heads were incubated for 5 min on a ThermoMixer (Eppendorf 5382000023), and 25 min in a rotating hybridization oven, and female heads were incubated for 10 min on the ThermoMixer and 50 min in rotating oven, all at 37°C. Heads were then rinsed once and fixed in 4% paraformaldehyde, 1X Ca$^{+2}$, Mg$^{+2}$ free PBS, and 0.25% Triton X-100 for 24 hr at room temperature on a rotator. All subsequent 4°C steps used a nutator, and room temperature steps used a rotator. Heads were washed for 30 min at room temperature at least three times in 1X PBS with 0.03% Triton X-100 (0.03% PBT). Antennae were then dissected into 0.5-mL microfuge tubes and dehydrated in 80% methanol/20% DMSO for 1 hr at room temperature. Antennae were washed in 0.03% PBT for 30 min at room temperature, and blocked/permeabilized in 1X PBS, 1% DMSO (Sigma 472301), 5% normal goat serum, 4% Triton X-100 for 24 hr at 4°C. Antennae were washed for 30 min at least five times with 0.03% PBT, 1% DMSO at room temperature, and then moved to primary antibody in 1X PBS, 1% DMSO, 5% normal goat serum, 0.03% Triton X-100 for 72 hr at 4°C. Primary antibodies used were mouse anti-*Apocrypta bakeri* Orco monoclonal antibody #15B2 (1:50 dilution, gift of Joel Butterwick and Vanessa Ruta), and rabbit anti-RFP (1:100, Rockland 600-401-379). Orco monoclonal antibody specificity was verified in *Aedes aegypti* by staining *orco* mutant antennae, which showed no staining (data not shown). Antennae were washed for 30 min at least five times with 0.03% PBT, 1% DMSO at room temperature, and then washed overnight in the same solution. Antennae were then moved to secondary antibody (1:200) and DAPI (1:10000, Sigma D9542) in 1X PBS, 1% DMSO, 5% normal goat serum, 0.03% Triton X-100 for 72 hr at 4°C. Secondary antibodies used were goat anti-mouse Alexa Fluor 488 (Thermo A-11001) and goat anti-rabbit Alexa Fluor 555 Plus (Thermo A32732). Antennae were washed for 30 min at least five times with 0.03% PBT, 1% DMSO at room temperature, and then washed overnight in the same solution. Antennae were rinsed in 1X PBS, rinsed three times in Slowfade Diamond (Thermo S36972), and mounted in Slowfade Diamond.

## Confocal image acquisition

Images were acquired with a Zeiss Axio Observer Z1 Inverted LSM 880 NLO laser scanning confocal microscope (Zeiss) with either 25x/0.8 NA (whole brains) or 40x/1.4 NA (antennal lobes, antennae)

immersion-corrected objective at a resolution of 1024 × 1024 or 2048 × 2048 (brains) or 3024 × 1024 (antennae) pixels. Confocal images were processed in ImageJ (NIH).

## Arm-next-to-cage assay

This assay was performed as described previously (*DeGennaro et al., 2013*). Briefly, for each trial, 20 adult mosquitoes were sorted under cold anesthesia (4°C) and placed in a cage and allowed to acclimate for 30 min. A human arm was placed 2.5 cm from one side of a standard 28 × 28×28 cm cage. Mosquitoes could not directly contact the human arm. A Logitech C920s HD Pro Webcam was positioned to take images of mosquitoes responding to the human arm. Trials ran for 10 min and images were acquired at a rate of 1 frame/s. To quantify mosquito responses, we manually counted the number of mosquitoes resting on the lower portion of the screen closest to the human arm.

## Feeding assay

Mosquitoes were cold-anesthetized, and 20 mosquitoes were sorted into 250 mL bottles covered with a taut net secured with rubber bands. They were allowed to acclimate for 24 hr with access to water through cotton balls. The following meals were presented: water, 10% sucrose, or sheep's blood (Hemostat DSB100) supplemented with 1 mM ATP (Sigma A6419). Meals were warmed to 45° C before being used in the assay. 10 mL of a given meal was pipetted into the bottle caps, animals were activated with a 30 s pulse of 4% $CO_2$, and bottles were inverted on top of the caps. Mosquitoes were allowed to feed on each meal through the net for 10 min and were then anesthetized at 4° C and scored as fed if any level of feeding was observed, as assessed by visual inspection of the abdomen of the animal.

## Insemination assay

Mosquitoes were separated by sex at the pupal stage and sex was confirmed within 24 hr of eclosion. For each trial, 10 female Liverpool-IB12 virgin mosquitoes were crossed to 11 virgin male mosquitoes of either Liverpool-IB12 or *fruitless*$^{\Delta M}$/*fruitless*$^{\Delta M\text{-}tdTomato}$ genotype in a bucket cage for 24 hr, with access to 10% sucrose. Mosquitoes were then anesthetized at 4°C, females separated from males, and female spermathecae were dissected to score for insemination as a sign of successful mating (*Degner and Harrington, 2016*). Control virgin females were dissected in parallel to verify absence of insemination.

## Quattroport olfactometer

Details of Quattroport fabrication and operation are available at https://github.com/VosshallLab/Basrur_Vosshall2020. Briefly, the Quattroport consists of four tubes, each with its own stimulus box, trap, and mosquito start chamber. There are adjustable gates between each chamber. The stimulus was placed upstream of a trap, and mosquitoes are prevented from contacting the stimulus by a mesh barrier. In each trial, four stimuli were run in parallel, with the positions of stimuli randomized and rotated between each trial. Air was filtered and pumped into each box, either at a final concentration of 1% $CO_2$ (for host-seeking assays) or at ambient $CO_2$ (honey-seeking assays). For all assays, ~20 mosquitoes were sorted and placed into canisters the day of behavior. Mosquitoes were allowed to acclimate in the assay for 10 min, then exposed to the stimulus for 30 s, after which gates were opened and animals allowed to fly for 5 min. After this time, gates were closed and mosquitoes were counted to quantify the percent of mosquitoes in the trap.

For honey assays, 3- to 7-day-old mosquitoes were fasted for 24 hr before the experiment by replacing 10% sucrose with a water source. $CO_2$ was not added for honey assays. Either 1 g of leatherwood honey (Tasmanian Honey Company, Tasmania, Australia) or glycerol (Sigma G5516) was applied to a 55 mm diameter Whatman filter paper circle (GE Healthcare, Buckinghamshire, UK) and placed in a Petri dish.

For host-seeking assays, mosquitoes were allowed access to sucrose before the experiment. A final concentration of 1% $CO_2$ was supplied in the airstream for the duration of the 5 min 30 s assay in all host-seeking assays (for both human forearm/odor stimuli and blank/unworn nylon controls). For live human host-seeking assays, a human subject placed their forearm on an acrylic box, exposing a 2.5 × 5 cm rectangle of skin to the airstream.

For human odor host-seeking assays, the same human subject wore a tan nylon sleeve (L'eggs Women's Comfortable Everyday Knee Highs Reinforced Toe Panty Hose, modified with scissors to remove the toe area) on their forearm. A second black nylon sleeve was worn on top of the tan nylon odor sampling sleeve to protect it from external odors. After 6 hr of continuous wear, the black nylon sleeve was discarded, and the tan nylon sleeve was frozen at −20°C. Nylons were used within 1 week of being worn. On the day of the assay, nylons were thawed for at least 1 hr at room temperature. A 10 × 14 cm piece of the sleeve was presented with the skin-contacting surface facing upward in the stimulus box along with $CO_2$. Unworn nylons were similarly frozen, thawed, and cut to serve as negative controls.

### Heat-seeking assay

Experiments were performed as previously described (*Corfas and Vosshall, 2015*; *McMeniman et al., 2014*). Briefly, 45–50 mosquitoes were fasted for 3 hr before the experiment by replacing 10% sucrose with a water source and were then transferred into a custom-made Plexiglass box (30 × 30×30 cm), with carbon-filtered air pumped continuously into the box via a diffusion pad installed on the ceiling of the enclosure. All stimulus periods lasted 3 min and were presented on a single Peltier element (6 × 9 cm, Tellurex) covered with a piece of standard white letter-size printer paper (NMP1120, Navigator) cut to 15 × 17 cm and held taut by a magnetic frame. $CO_2$ pulses (20 s, to >1000 ppm above background levels) were added to the air stream and accompanied all stimulus period onsets. Mosquito landings on the Peltier were monitored by fixed cameras (FFMV-03M2M-CS, Point Grey Research) with images acquired at 1 Hz. Images were analyzed using custom MATLAB scripts to count mosquito landings within a fixed target region. Mosquito occupancy on the Peltier was quantified during seconds 90–180 of each stimulus period.

### Quantification, statistical analysis, and reproducibility

All statistical analyses were performed using GraphPad Prism Version 8. Data collected as percent of total are shown as mean ± s.e.m. Details of statistical methods are reported in the figure legends. Preliminary experiments were used to assess variance and determine sample sizes before carrying out experiments. Typically, sample sizes were n = 6–14 groups of 10-20 mosquitoes in behavioral assays. We used similar sample sizes for all experiments where the same variable was being compared. No data were excluded from this study. Since mosquito behavior is variable, all olfactometer experiments with a human arm were carried out repeatedly to assess the effect of external environmental conditions on behavior. No experiments were performed on days when < 40% of wild-type females responded to a live human arm. No data met these exclusion criteria. All attempts at replication over multiple days were successful. We carried out all experiments with different groups of animals hatched up to 4 weeks apart, and over multiple days. Several experiments were carried out repeatedly over the course of this study, namely the wild-type female response to a live human arm in Quattroport olfactometer assays. These results were robust and reliable over the course of the many years it took to complete this study. For all experiments, mosquitoes from a cage were randomly selected and sorted by sex into groups for behavioral assays. All stimuli and genotypes were interspersed, and positions were randomized when possible. Every experiment involved replicates collected over multiple days, to ensure that there was no effect of daily environmental or experimental conditions. We also collected a similar sample size for each variable every time the experiment was run, to ensure no effect of external conditions. Blinding to genotype was performed in the heat-seeking assays. The experimenter was not blinded to genotype in the host-seeking assays because the mutant phenotype we describe is so robust it was impossible to be blinded in these assays.

### Data and software availability

All raw data are provided in the accompanying Source Data files. Plasmids are available at Addgene (#141099, #141100). RNA-seq data are available in the Short Read Archive at Genbank (Bioproject: PRJNA612100). Details of Quattroport fabrication and operation are available at Github: https://github.com/VosshallLab/Basrur_Vosshall2020.

## Acknowledgements

We thank Richard Benton, Josie Clowney, Lindy McBride, Vanessa Ruta, David Stern, Nilay Yapici, and members of the Vosshall Lab for comments on the manuscript; Gloria Gordon and Libby Mejia for expert mosquito rearing; Joel Butterwick and Vanessa Ruta for sharing unpublished anti-*Apocrypta bakeri* Orco monoclonal antibodies; the colleagues who shared mosquito species with us [William Bradshaw and Christina Holzapfel (*Wyeomyia smithii*); Larry Zwiebel and Jason Pitts (*Toxorhynchites amboinensis*); Flaminia Catteruccia (*Anopheles gambiae*)] and those who shared insect photos with us [Alex Wild (*Aedes aegypti*, *Anopheles gambiae*), Lawrence Reeves of Florida Medical Entomology Laboratory (*Wyeomyia smithii*, *Toxorhynchites rutilus*, *Culex salinarius*), Francisco Romero, Veronica Corrales-Carvajal, Carlos Ribeiro, and Nicolas Gompel (*Drosophila melanogaster*)]; Erich Jarvis, Vanessa Ruta, and Li Zhao for advice and discussion; Ben Matthews and Zach Gilbert for assistance in designing and testing of sgRNAs and advice on the CRISPR protocol; Ben Matthews for advice on bioinformatics; Rob A Harrell II at the Insect Transgenesis Facility at the University of Maryland for CRISPR-Cas9 injections; Jim Petrillo and Kunal Shah at the Rockefeller Precision Instrument Technology center for help in development and construction of the Quattroport assay; Meg Younger for advice on brain dissections and antibody staining; Ella Jacobs for training in spermathecae dissections; Christina Pyrgaki, Carlos Rico, and Alison North at the Rockefeller Bio-Imaging Resource Center for assistance with confocal imaging; Connie Zhao at the Rockefeller Genomics Resource Center for advice on RNA-seq library preparation and sequencing; Román Corfas for assistance with the heat-seeking assay.

## Additional information

### Funding

| Funder | Grant reference number | Author |
| --- | --- | --- |
| Howard Hughes Medical Institute | Vosshall-Investigator | Leslie B Vosshall |
| National Center for Advancing Translational Sciences | UL1 TR000043 | Leslie B Vosshall |
| Harvey L Karp Discovery Award | Postdoctoral fellowship | Maria Elena De Obaldia Takeshi Morita |
| Japan Society for the Promotion of Science | JSPS Overseas Research Fellowship | Takeshi Morita |
| Helen Hay Whitney Foundation | HHW Fellowship | Maria Elena De Obaldia |
| National Center for Advancing Translational Sciences | UL1 TR001866 | Maria Elena De Obaldia |
| National Institute on Deafness and Other Communication Disorders | F30DC017658 | Margaret Herre |
| National Institute of General Medical Sciences | T32GM007739 | Margaret Herre |
| Quadrivium Foundation | Vosshall/Herre-Investigator | Margaret Herre |

The funders had no role in study design, data collection and interpretation, or the decision to submit the work for publication.

### Author contributions

Nipun S Basrur, Conceptualization, Data curation, Software, Formal analysis, Validation, Investigation, Visualization, Methodology, Writing - original draft, Writing - review and editing; Maria Elena De Obaldia, developed Quattroport assay, Methodology, Writing - review and editing; Takeshi Morita, Conceptualization, Data curation, Formal analysis, Investigation, Writing - review and editing; Margaret Herre, Methodology, Writing - review and editing; Ricarda K von Heynitz, Yael N Tsitohay, Investigation, Methodology, Writing - review and editing; Leslie B Vosshall, Conceptualization, Supervision, Funding acquisition, Project administration, Writing - review and editing

Author ORCIDs
Nipun S Basrur  https://orcid.org/0000-0002-7068-7798
Maria Elena De Obaldia  https://orcid.org/0000-0003-2488-3672
Takeshi Morita  https://orcid.org/0000-0002-8570-6744
Margaret Herre  https://orcid.org/0000-0001-7868-3321
Ricarda K von Heynitz  https://orcid.org/0000-0002-3038-3036
Yael N Tsitohay  http://orcid.org/0000-0002-8716-9444
Leslie B Vosshall  https://orcid.org/0000-0002-6060-8099

## Ethics

Human subjects: Blood-feeding procedures and behavioral experiments with human volunteers were approved and monitored by The Rockefeller University Institutional Review Board (IRB protocol LV-0652). Human subjects gave their written informed consent to participate.
Animal experimentation: Blood-feeding procedures with live mice were approved and monitored by The Rockefeller University Institutional Animal Care and Use Committee (IACUC protocol 17018).

## Decision letter and Author response

Decision letter https://doi.org/10.7554/eLife.63982.sa1
Author response https://doi.org/10.7554/eLife.63982.sa2

## Additional files

### Supplementary files

- Supplementary file 1. Primer sequences used to clone plasmids.
- Supplementary file 2. List of successful and attempted genetic manipulations of the *fruitless* locus.
- Transparent reporting form

### Data availability

All raw data are provided in the accompanying source data files. Plasmids are available at Addgene (#141099, #141100). RNA-seq data are available in the Short Read Archive at Genbank (Bioproject: PRJNA612100). Details of Quattroport fabrication and operation are available at Github: https://github.com/VosshallLab/Basrur_Vosshall2020 (copy archived at https://archive.softwareheritage.org/swh:1:rev:529b0ba393bac09207d298e73e425b452338e876).

The following dataset was generated:

| Author(s) | Year | Dataset title | Dataset URL | Database and Identifier |
|---|---|---|---|---|
| Basrur N | 2020 | Sex-specific mosquito brain transcriptomes | https://www.ncbi.nlm.nih.gov/bioproject/?term=PRJNA612100 | NCBI BioProject, PRJNA612100 |

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
