## [Decision Letter]

**Acceptance summary:**

This study presents the remarkable observation that *fruitless* mutant males *Aedes aegypti* mosquitoes acquire aspects of human host-seeking, a behavior normally seen only in females. This is a novel and exciting discovery that suggests that *fruitless* functions in the male olfactory system to repress a key component of host seeking. The study provides a solid foundation for examining the neural basis for sexual dimorphism in human host-seeking behavior in mosquitoes.

**Decision letter after peer review:**

Thank you for submitting your article "*fruitless* mutant male mosquitoes gain attraction to human odor" for consideration by *eLife*. Your article has been reviewed by three peer reviewers, and the evaluation has been overseen by Kristin Scott as Reviewing Editor and Catherine Dulac as the Senior Editor. The reviewers have opted to remain anonymous.

The reviewers have discussed the reviews with one another and the Reviewing Editor has drafted this decision to help you prepare a revised submission.

Summary:

In this work Basrur et al. present the remarkable observation that *fruitless* mutant males *Aedes aegypti* mosquitoes acquire aspects of human host-seeking, a behavior normally seen only in females. Unlike their wild-type controls, *fruitless* mutant male mosquitoes are attracted to human odor. Interestingly, *fruitless* males do not also acquire heat seeking, a second component of the female-specific human host-seeking program, and do not feed on blood. These results suggest that *fruitless* functions in the male olfactory system to repress a key component of host seeking.

The reviewers are in agreement that the work is of very high quality, the experiments are well designed and analyzed, the figures are of excellent quality and the paper well written. Textual clarifications are requested below.

Essential revisions:

1) In the Discussion, the authors provide two different models to explain the function of *fruitless* in suppressing host seeking behavior in male mosquitos. In one model, *fruitless* modulates the detection of human odors in the periphery. In another, *fruitless* functions in the central brain to reroute olfactory information. The authors favor the latter and dismiss the first based on their observation that the *fruitless* mutation does not change the number or position of the *fruitless*-positive glomeruli in the antennal lobe. This argument is unconvincing. Is it possible that *fruitless* suppresses the expression of specific olfactory receptor genes? Does the *fruitless* mutation change glomerular volume? Changes in OR expression may alter olfactory tuning and volume changes may alter olfactory response intensity. Without functional data, it would be difficult to favor either model. It is interesting that *fruitless* males do not heat seek, do not feed on blood and are not feminized in their response to CO2 (is this correct?) indicating that *fruitless* controls an important but delimited part of host seeking behavior. This Discussion should be revised. Relatedly, the Abstract sentence "suggest[ing] that male mosquitoes possess the neural circuits required to host-seek and that removing *fruitless* reveals this latent behavior in males" might be revised.

2) The text is sometimes rather imprecise and could benefit from some refinement. Consider changing the following sections/sentences:

"*fruitless* encodes a BTB zinc-finger transcription factor that is thought to control cell identity and connectivity during development (Ito et al., 2016; Neville et al., 2014), as well as the functional properties of neurons in adulthood (Sethi et al., 2019), although its genomic targets and the molecular mechanism by which it acts remain unclear." Genomic targets and molecular mechanisms are not completely unclear! Neville et al., 2014 identifies targets, as does Vernes, 2014. There is substantial recent work on *fruitless* targets (lola, robo, chromatin modifiers), *fruitless* interaction partners and molecular mechanisms by the Yamamoto group- see e.g. the recent review from Sato and Yamamoto, 2020, as well as e.g. Sato et al. 2020, Commun Biol doi: 10.1038/s42003-020-01327-z. A follow up to Sethi et al., 2019 with more mechanistic insight is available at Cell reports Sneek Peek: https://papers.ssrn.com/sol3/papers.cfm?abstract_id=3624384

"However, it is unlikely that any Fruitless protein is stably expressed in the female. In *Drosophila*, female *fruitless*peptides are not detected." Specify: sex specific proteins/peptides, in the adult. In *Drosophila*, common isoforms of Fruitless are expressed in females during development. The uninitiated reader would benefit from a bit more detail about sex specific vs. common isoforms.

"The projections of *fruitless*>tdTomato neurons appear to be dramatically sexually dimorphic relative to *D. melanogaster*." Unclear: P1 fru transcripts, as judged by fru-GAL4 (Stockinger line) expression show a clear dimorphic expression in *Drosophila* – the level of "dimorphicness" in mosquitos does not seem "dramatically" more/stronger to me.

"If *fruitless* function in *Aedes aegypti* were limited to mating and courtship as it is in *Drosophila*". Fruitless also functions in aggression in *Drosophila*: Vrontou et al., 2006. Actually, here males mutant for FruM (so called FruF males) "gain" female specific behavior/fighting style. Maybe that would be worth mentioning in the Discussion as well – that in *Drosophila*, FruM also seems to suppress some female specific behaviors.

"Since males are able to host-seek in the absence of *fruitless*, the sex of the mosquito does not intrinsically regulate the development and function of brain circuits

controlling host-seeking behavior." Unclear – absence/presence of *fruitless* is an important aspect of "the sex" of the mosquito- or what else do the authors mean by "the sex"?

"Another possibility is that mosquitoes duplicated and co-opted the ancestral *fruitless*-expressing mating neural circuits, and retuned the inputs and outputs into this circuit to drive host-seeking. In these duplicated circuits,…". I have a hard time understanding the concept of a "duplicated mating circuit" which is subsequently retuned. I don't think the gene duplication-diversification model/concept can be so easily re-used for circuits. I'd appreciate if the authors could clarify what they exactly mean here and give some citations, if available.

3) The figures are not mentioned in order in the text- Figure 3 is referred to in the Results before the second part of Figure 2 (D-K, behavioral data). Please rearrange figures or text to make it fit.

4) The authors claim that the construct they use as marker and mutant allele *fruitless^∆M^*-CsChrimson:tdtomato lead to Chrimson expression that is too low for any optogenetic activation (subsection “*fruitless^∆M^andfruitless^∆M-tdTomato^* strain generation). Could they elaborate if and how they tested for the effect of a functional Chrimson on a behavioral level? I am convinced from Figure 4D that the Chrimson allele alone does not elicit attraction to human odor in males, but was there indication that it had an influence on other behaviors in either males or females? Alternatively, did the authors confirm that the homozygous deletion allele without Chrimson (*fruitless^∆M^*/*fruitless^∆M^*) gives the same phenotype?

5) Statistical analysis in Figure 4. It was stated in the legend "In C, E, data labeled with different letters are significantly different from each other (Kruskal-Wallis test with Dunn's multiple comparisons, p < 0.05)". First, this statement is likely referred to panels B and D, not C and E. Second, it is unclear whether statistical test was run between different temperatures in panel B. If so, the letters for 26 degree appear to be "b b b", instead of "a a a".

6) Given the careful analysis of the *fruitless* gene structure, it is very likely that the *fruitless^∆M^* mutation does not affect female mating or host-seeking behavior. However, mutant females were only examined with the human odor host-seeking assay (Figure 4D). Readers would be interested to see the experiments carried out as in Figure 5—figure supplement 2 to be performed with *fruitless^∆M^* mutant females. Such data would strengthen the argument that females do not make a full-length *fruitless* protein and the *fruitless^∆M^* mutation does not affect female mating or host-seeking behavior. If the authors already have this data, it would be a nice addition.

---

## [Author Response]

Essential revisions:1) In the Discussion, the authors provide two different models to explain the function of fruitless in suppressing host seeking behavior in male mosquitos. In one model, fruitless modulates the detection of human odors in the periphery. In another, fruitless functions in the central brain to reroute olfactory information. The authors favor the latter and dismiss the first based on their observation that the fruitless mutation does not change the number or position of the fruitless-positive glomeruli in the antennal lobe. This argument is unconvincing. Is it possible that fruitless suppresses the expression of specific olfactory receptor genes? Does the fruitless mutation change glomerular volume? Changes in OR expression may alter olfactory tuning and volume changes may alter olfactory response intensity. Without functional data, it would be difficult to favor either model. It is interesting that fruitless males do not heat seek, do not feed on blood and are not feminized in their response to CO2 (is this correct?) indicating that fruitless controls an important but delimited part of host seeking behavior. This Discussion should be revised. Relatedly, the Abstract sentence "suggest[ing] that male mosquitoes possess the neural circuits required to host-seek and that removing fruitless reveals this latent behavior in males" might be revised.

We have revised the Discussion and Abstract to place equal emphasis on both peripheral and central models and clarified the significant technical and genetic advances that are necessary to distinguish between the models.

2) The text is sometimes rather imprecise and could benefit from some refinement. Consider changing the following sections/sentences:"fruitless encodes a BTB zinc-finger transcription factor that is thought to control cell identity and connectivity during development (Ito et al., 2016; Neville et al., 2014), as well as the functional properties of neurons in adulthood (Sethi et al., 2019), although its genomic targets and the molecular mechanism by which it acts remain unclear." Genomic targets and molecular mechanisms are not completely unclear! Neville et al. 2014 identifies targets, as does Vernes, 2014. There is substantial recent work on fruitless targets (lola, robo, chromatin modifiers), fruitless interaction partners and molecular mechanisms by the Yamamoto group- see e.g. the recent review from Sato and Yamamoto, 2020, as well as e.g. Sato et al. 2020, Commun Biol doi: 10.1038/s42003-020-01327-z. A follow up to Sethi et al., 2019 with more mechanistic insight is available at Cell reports Sneek Peek: https://papers.ssrn.com/sol3/papers.cfm?abstract_id=3624384

We apologize for this oversight our imperfect scholarship and have revised this sentence to include the suggested citations on the targets and mechanism of action of *fruitless*.

"However, it is unlikely that any Fruitless protein is stably expressed in the female. In Drosophila, female fruitless peptides are not detected." Specify: sex specific proteins/peptides, in the adult. In *Drosophila,* common isoforms of Fruitless are expressed in females during development. The uninitiated reader would benefit from a bit more detail about sex specific vs. common isoforms.

We have revised the text to clarify that *fruitless* has broadly expressed common isoforms and neuronally enriched sex-specific isoforms and mentioned that the latter are not likely to lead to functional protein in females.

"The projections of fruitless>tdTomato neurons appear to be dramatically sexually dimorphic relative to *D. melanogaster*." Unclear: P1 fru transcripts, as judged by fru-GAL4 (Stockinger line) expression show a clear dimorphic expression in Drosophila – the level of "dimorphicness" in mosquitos does not seem "dramatically" more/stronger to me.

We agree that our original statement was subjective and have revised this sentence to remove comparisons with *Drosophila*.

"If fruitless function in Aedes aegypti were limited to mating and courtship as it is in Drosophila". Fruitless also functions in aggression in *Drosophila:* Vrontou et al. 2006. Actually, here males mutant for FruM (so called FruF males) "gain" female specific behavior/fighting style. Maybe that would be worth mentioning in the Discussion as well – that in Drosophila, FruM also seems to suppress some female specific behaviors.

We thank the reviewers for highlighting the important research on sex-specific *Drosophila* aggression and have now included this reference in the Discussion and relevant portions of the Results and Introduction.

"Since males are able to host-seek in the absence of fruitless, the sex of the mosquito does not intrinsically regulate the development and function of brain circuitscontrolling host-seeking behavior." Unclear – absence/presence of fruitless is an important aspect of "the sex" of the mosquito- or what else do the authors mean by "the sex"?

We have revised this sentence to clarify that other genes in the sex-determination hierarchy do not control sexual dimorphism in host-seeking behavior.

"Another possibility is that mosquitoes duplicated and co-opted the ancestral fruitless-expressing mating neural circuits, and retuned the inputs and outputs into this circuit to drive host-seeking. In these duplicated circuits,…". I have a hard time understanding the concept of a "duplicated mating circuit" which is subsequently retuned. I don't think the gene duplication-diversification model/concept can be so easily re-used for circuits. I'd appreciate if the authors could clarify what they exactly mean here and give some citations, if available.

We have revised this paragraph to include a clarification of this concept, and included relevant citations to highlight the rich scholarship on the mechanisms of neural circuit evolution, namely recent work on the role of circuit duplication in the evolution of vocal learning and of cerebellar nuclei.

3) The figures are not mentioned in order in the text- Figure 3 is referred to in the Results before the second part of Figure 2 (D-K, behavioral data). Please rearrange figures or text to make it fit.

We have revised the figures to appear in the order in which they are referred to in the text. Specifically, we have expanded Figure 2 to exclusively show *fruitless* expression in the central brain, and now have a new Figure 4 with the behavioral experiments formerly in Figure 2.

4) The authors claim that the construct they use as marker and mutant allele fruitless^∆M^-CsChrimson:tdtomato lead to Chrimson expression that is too low for any optogenetic activation (subsection “fruitless^∆M^ and fruitless^∆M-tdTomato^ strain generation). Could they elaborate if and how they tested for the effect of a functional Chrimson on a behavioral level? I am convinced from Figure 4D that the Chrimson allele alone does not elicit attraction to human odor in males, but was there indication that it had an influence on other behaviors in either males or females? Alternatively, did the authors confirm that the homozygous deletion allele without Chrimson (fruitless^ΔM^/fruitless^∆M^) gives the same phenotype?

Potential CsChrimson driven activation of fruitless+ neurons is an important concern to rule out. We note that CsChrimson:tdTomato expression is very weak since it is driven by the endogenous *fruitless* promoters without any amplification from QF2. In support of this, CsChrimson:tdTomato endogenous fluorescence was not visible under a confocal microscope, even at high laser intensities, and required amplification from antibodies in all our images. We also note that animals were not fed trans-retinal, the necessary cofactor for CsChrimson activity. Even when we fed animals retinal and attempted to substitute human odor and CO_2_ with optogenetic activation of *fruitless*+ neurons in a blood-feeding assay, we did not observe increased feeding or any other behaviors, likely due to this weak expression (preliminary data not shown). Finally, we have now included more behavioral data in Figure 5—figure supplement 1, showing that *fruitless^∆M-CsChrimson-tdTomato/+^* males and females are able to mate normally, and *fruitless^∆M-CsChrimson-tdTomato/+^*females show normal blood-feeding and egg-laying behavior. We have not examined *fruitless^∆M /∆M^* animals due to the difficulty of obtaining these animals without molecular genotyping of each individual. However, given the weak expression of CsChrimson and the robust behavior of heterozygote animals, we consider it unlikely that this allele is significantly affecting mosquito behavior. We have now included these details in the Materials and methods section.

5) Statistical analysis in Figure 4. It was stated in the legend "In C, E, data labeled with different letters are significantly different from each other (Kruskal-Wallis test with Dunn's multiple comparisons, p < 0.05)". First, this statement is likely referred to panels B and D, not C and E. Second, it is unclear whether statistical test was run between different temperatures in panel B. If so, the letters for 26 degree appear to be "b b b", instead of "a a a".

We thank the reviewers for noting the error in the legend, which we have now corrected. The test was run within genotypes for a given temperature, and not between temperatures. The figure legends for this panel, as well as other ambiguous panels in Figure 4 have been updated to clarify this point.

6) Given the careful analysis of the fruitless gene structure, it is very likely that the fruitless-deltaM mutation does not affect female mating or host-seeking behavior. However, mutant females were only examined with the human odor host-seeking assay (Figure 4D). Readers would be interested to see the experiments carried out as in Figure 5—figure supplement 2 to be performed with fruitless^∆M^ mutant females. Such data would strengthen the argument that females do not make a full-length fruitless protein and the fruitless^∆M^ mutation does not affect female mating or host-seeking behavior. If the authors already have this data, it would be a nice addition.

We have now added Figure 5—figure supplement 1 showing blood-feeding, egg-laying, and mating status of *fruitless^∆M/∆M-CsChrimson-tdTomato^* females, in addition to heterozygous controls, as well as wild-type females crossed to various male genotypes. Notably, *fruitless^∆M/∆M-CsChrimson-tdTomato^* females blood-feed at comparable rates to wild-type females (Figure 5—figure supplement 1A), are able to lay eggs at high rates (Figure 5—figure supplement 1B), and are able to successfully mate with wild-type males (Figure 5—figure supplement 1C), suggesting that the *fruitless^∆M^*mutation does not significantly affect female mating, host-seeking, or feeding behavior. These data support the model where sex-specific *fruitless* protein is not functional in females.